# Early rehabilitation to prevent postintensive care syndrome in patients with critical illness: a systematic review and meta-analysis

Ryota Fuke,[1] Toru Hifumi,[2] Yutaka Kondo,[3] Junji Hatakeyama,[4] Tetsuhiro Takei,[5] Kazuma Yamakawa,[5] Shigeaki Inoue,[6] Osamu Nishida[7]

RF and TH contributed equally.

For numbered affiliations see end of article.

**Correspondence to**
Dr Shigeaki Inoue;
caf55000@gmail.com

## ABSTRACT

**Introduction** We examined the effectiveness of early rehabilitation for the prevention of postintensive care syndrome (PICS), characterised by an impaired physical, cognitive or mental health status, among survivors of critical illness.

**Methods** We performed a systematic literature search of several databases (Medline, Embase and Cochrane Central Register of Controlled Trials) and a manual search to identify randomised controlled trials (RCTs) comparing the effectiveness of early rehabilitation versus no early rehabilitation or standard care for the prevention of PICS. The primary outcomes were short-term physical-related, cognitive-related and mental health-related outcomes assessed during hospitalisation. The secondary outcomes were the standardised, long-term health-related quality of life scores (EuroQol 5 Dimension (EQ5D) and the Medical Outcomes Study 36-Item Short Form Health Survey Physical Function Scale (SF-36 PF)). We used the Grading of Recommendations Assessment, Development and Evaluation approach to rate the quality of evidence (QoE).

**Results** Six RCTs selected from 5105 screened abstracts were included. Early rehabilitation significantly improved short-term physical-related outcomes, as indicated by an increased Medical Research Council scale score (standardised mean difference (SMD): 0.38, 95% CI 0.10 to 0.66, p=0.009) (QoE: low) and a decreased incidence of intensive care unit-acquired weakness (OR 0.42, 95% CI 0.22 to 0.82, p=0.01, QoE: low), compared with standard care or no early rehabilitation. However, the two groups did not differ in terms of cognitive-related delirium-free days (SMD: −0.02, 95% CI −0.23 to 0.20, QoE: low) and the mental health-related Hospital Anxiety and Depression Scale score (OR: 0.79, 95% CI 0.29 to 2.12, QoE: low). Early rehabilitation did not improve the long-term outcomes of PICS as characterised by EQ5D and SF-36 PF.

**Conclusions** Early rehabilitation improved only short-term physical-related outcomes in patients with critical illness. Additional large RCTs are needed.

For numbered affiliations see end of article.

## Strengths and limitations of this study

► This meta-analysis is the first meta-analysis of comprehensive postintensive care syndrome (PICS) based on randomised controlled trials (RCTs), in which the study intervention populations were limited to early rehabilitation.
► This meta-analysis was mainly limited by (1) the small number of patients in the included RCTs, (2) a lack of detailed analysis of the adverse effects and PICS domains after hospital discharge and (3) no confirmation of the exact first day of rehabilitation initiation in one of the six included RCTs.
► We used the Grading of Recommendations Assessment, Development and Evaluation approach in the review process.

have reduced the mortality of patients with critical illness over the past four decades.[1] However, this evolution of life-saving interventions has led to increasing numbers of surviving critically ill patients with an impaired ability to return to physical and mental aspects of normal life.[2] These persistent physical, cognitive and mental impairments experienced by ICU survivors present serious obstacles to discharge from the hospital to a home setting and, once home, to a return to normal daily life.[2 3]

Postintensive care syndrome (PICS) was established as a new syndrome encompassing new or worsening impairments in physical, cognitive or mental health status that arise after critical illness and persist beyond acute care hospitalisation, with the aim initiating improvements for ICU survivors and their families across the continuum of care.[4] Since the establishment of PICS in 2010, observational studies have evaluated the independent factors associated with this syndrome[5]; however, few intervention studies have targeted the prevention of PICS.

## INTRODUCTION

Dramatic developments and improvements in the technique, instruments and education systems used in intensive care units (ICUs)

Physiotherapy with early rehabilitation is considered an integral component of the multidisciplinary management of patients in ICUs. In other populations, exercise has been shown to improve strength and function, decrease inflammation[6–8] and affect oxidative stress,[9–12] leading to suggestions that early physiotherapy may prevent or reverse some physical impairments in ICU patients. However, no systematic review has investigated the effectiveness of early rehabilitation for the prevention of PICS in ICU patients. The present systematic review aimed to assess the effectiveness of early rehabilitative interventions for the prevention of PICS in ICU patients.

## METHODS

### Protocols and registration

This review protocol was registered in PROSPERO, an International Prospective Register of Systematic Reviews at the National Institute for Health Research and Centre for Reviews and Dissemination at the University of York (http://www.crd.york.ac.uk/PROSPERO/) (Registration No. CRD42016039759).[13] This protocol adhered to the Preferred Reporting Items for Systematic Reviews and Meta-Analyses Protocols (PRISMA-P) statements,[14 15] and the systematic review was reported according to PRISMA guidelines.[14 16 17] The protocol for this systematic review was previously published.[18]

### Search strategy

We searched the Medline (via PubMed from 1996 to 7 June 2016), Embase (until 7 June 2016) and Cochrane Central Register of Controlled Trials (CENTRAL) databases (until 7 June 2016) for full-text clinical trials conducted in humans to retrieve relevant articles for the literature review. We also performed a manual search to retrieve all potentially relevant articles from 8 June to 31 August 2016. The key search terms used to identify potential studies are listed in online supplementary file 1. Finally, we updated our search in Medline (via PubMed from 8 June 2016 to 15 January 2018) using the same key search terms.

### Study selection and inclusion criteria

Three authors (YK, RF, SI) performed the first-line comprehensive literature search and removal of duplicates. Subsequently, two randomly selected authors independently screened the study titles and abstracts for potential relevance. The kappa value for selection bias was 0.396. When disagreements arose between reviewers, the full text of the paper was retrieved, and the disagreements were reconsidered and discussed until a consensus was reached. The full texts of articles included in the final selection were reviewed by two randomly chosen authors. We included trials with the following characteristics:

*Study types*: Randomised controlled trials (RCTs) were included; non-randomised and observational studies were excluded.

*Population*: Adult patients (aged >18 years) admitted to the ICU were included. We excluded patients with traumatic brain injury and stroke who did not fulfil the PICS criteria, specifically those with acquired physical and psychiatric/cognitive dysfunction after ICU admission.

*Intervention*: The intervention was early rehabilitation, defined as (1) starting at an earlier time point than usual care or (2) administration within 7 days of ICU admission. 'Rehabilitation' included all physiotherapy, occupational therapy and palliative care-related support. We excluded RCTs in which rehabilitation was initiated before ICU admission and those comparing early rehabilitation with another intervention.

*Control*: The control group received standard care or no early rehabilitation.

*Outcome*: The primary outcomes (ie, short-term outcomes assessed during hospitalisation) were as follows: (1) physical-related outcomes (incidence of ICU-acquired weakness (AW), Medical Research Council (MRC)[19] scale score), (2) cognitive-related outcomes (delirium-free days) and (3) mental status-related outcomes (Hospital Anxiety and Depression Scale (HADS)).[20] The secondary outcomes (long-term outcomes assessed postdischarge) included the standardised Health-Related Quality of Life with EuroQol 5 Dimensions (EQ5D) test[21 22] and the Medical Outcomes Study 36-Item Short Form Health Survey Physical Function scale (SF-36 PF)[23] as measures of long-term physical function.

### Assessment of the risk of bias

To assess the quality of the included studies, we adopted the Cochrane risk of bias tool.[24] Two randomly selected authors performed the risk of bias assessment; disagreements were resolved by discussion, with the inclusion of a third reviewer if necessary. We considered the risk of bias for each element to be 'high' when bias was present and likely to affect outcomes, and 'low' when bias was not present or present but unlikely to affect outcomes.[25]

### Rating the quality of evidence using the Grading of Recommendations Assessment, Development and Evaluation approach

Two authors (RF and TH) independently used the Grading of Recommendations Assessment, Development, and Evaluation (GRADE) tool to rate the quality of evidence (QoE). We used the GRADE approach to rate the QoE of early rehabilitation for outcomes. Although the QoE represents a continuum, we assessed the quality of the body of evidence for each outcome categorised as high, moderate, low or very low using the GRADE pro Guideline Development Tool.

### Data extraction and management

The author(s), title, journal name, year of publication, website URL and abstract of each included article were identified. Conference abstracts were excluded. Data from each study were extracted independently by two review authors. When faced with insufficient or incomplete data,

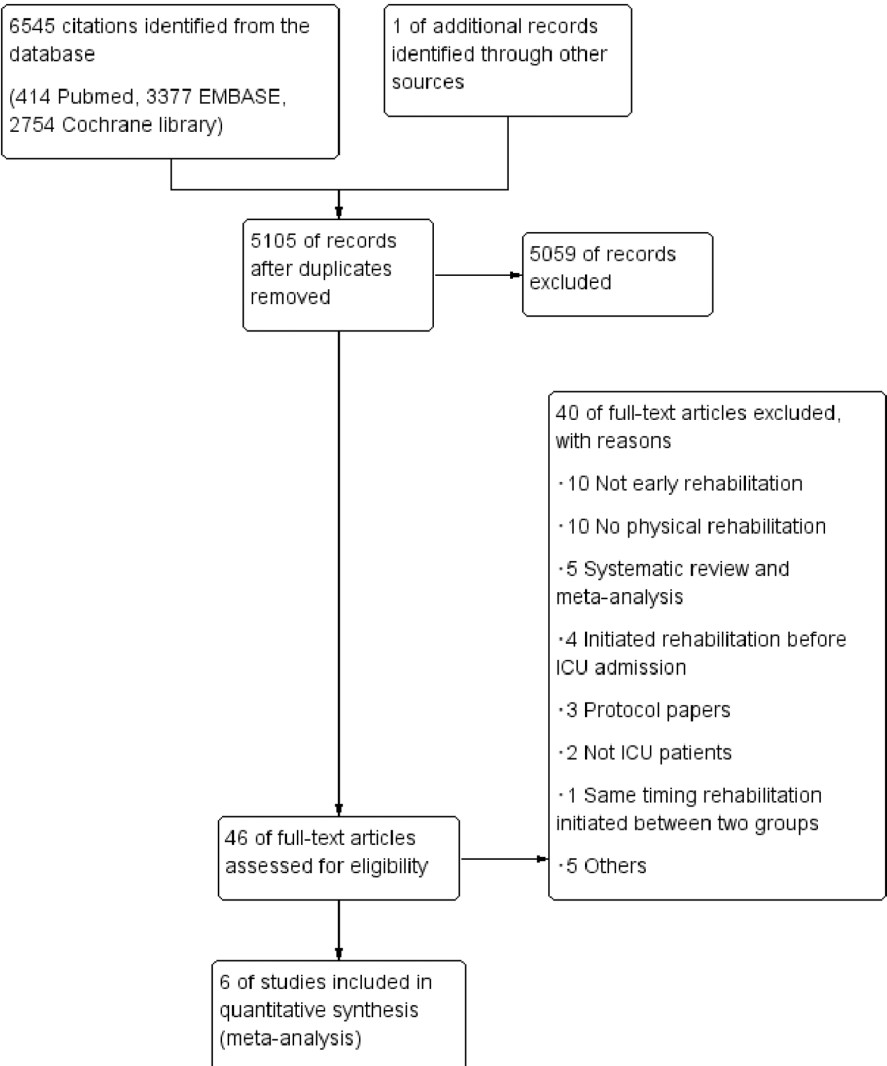

**Figure 1** Flow chart of the process of study identification and inclusion. ICU, intensive care unit.

we contacted the authors of the studies directly. Information related to the study population, sample size, interventions, comparators, potential biases in the conduct of the trial, outcomes (including adverse events), follow-up and statistical analysis methods were extracted from the original reports into specially designed paper forms and then entered into a spreadsheet.

### Summarisation of data and treatment effects

As data from several trials were available, we performed a meta-analysis according to the Cochrane Handbook for Systematic Reviews of Interventions and PRISMA guidelines using Review Manager software (RevMan V.5.3, Copenhagen, Denmark: The Nordic Cochrane Centre, the Cochrane Collaboration 2014). We summarised the results of this meta-analysis using the generic inverse variance method to facilitate the pooling of estimates of treatment effects. We used ORs with 95% CIs for dichotomous outcomes, and mean differences or standardised mean differences (SMDs) with 95% CIs for continuous outcomes when appropriate. If a quantitative synthesis

was not appropriate for a particular outcome, we devised a qualitative summary of that outcome.

### Assessment of heterogeneity

For each outcome, we assessed heterogeneity between trials using $I^2$ statistics to quantify inconsistencies. We considered that significant heterogeneity was present when the reason for heterogeneity could not be explained and the $I^2$ was ≥50%. The presence of strong clinical heterogeneity was considered when deciding to perform a quantitative data synthesis or a sensitivity analysis with a special focus.[26]

### Data synthesis

Estimates were pooled using a random effects model. We performed our analysis based on all published or available data.

### RESULTS
### Literature search

The PRISMA flow chart of study selection is shown in figure 1. The combined search strategy identified 5105

citations, of which 46 were judged to be potentially eligible based on the abstract. After excluding 40 of these citations following a full-text review (online supplementary file 2), six RCTs that had enrolled 709 patients were deemed eligible and were included in the review.[27–32] The updated literature search in Medline also identified a total of 320 citations, but no studies were judged to be potentially eligible based on the inclusion criteria.

## Characteristics of included studies

The study characteristics are summarised in table 1. Early rehabilitation was initiated within 3 days (median or mean) of ICU admission in five studies.[27 29–32] The first day of rehabilitation was not confirmed in three studies, although early rehabilitation was described in the text, and we contacted the authors to acquire additional information.[28] The rehabilitation strategies administered to each intervention group varied among protocols.

## Risk of bias in the included trials

The evaluation of the risk of bias in each included RCT is shown in the risk of bias summary for PICS-related outcomes without in figure 2. Regarding blinding, six trials involved open-label RCTs. The blinding of outcome assessments was described for all RCTs. In all RCTs, attrition bias was observed because of a per-protocol analysis, the inclusion of many patients who dropped out of the study and death. The performance bias of mortality as an outcome measure was low, however, because blinding did not affect mortality.

## Outcomes

The quality assessment and a summary of each outcome measure are shown in table 2.

*GRADE working group grades of evidence: High quality*—further research is very unlikely to change our confidence in the estimate of effect. Moderate quality—further research is likely to have an important impact on our confidence in the estimate of effect and may change the estimate. Low quality—further research is very likely to have an important impact on our confidence in the estimate of effect and is likely to change the estimate. Very low quality—we are very uncertain about the estimate.

### Primary outcomes (short-term outcomes)
#### Physical-related outcomes

*Incidence of ICU-AW*: Two RCTs reported the incidence of ICU-AW (figure 3). There was a significant reduction in the incidence of ICU-AW in the early rehabilitation group according to the random effects model (two trials, n=154; OR 0.42, 95% CI 0.22 to 0.82, p=0.01; $I^2=0\%$).

The risk of bias was determined to be serious because all RCTs included a moderate risk of bias. Imprecision was also determined to be serious because of the small number of included patients as per the optimal information size (OIS) criteria. Therefore, the QoE was

downgraded from high to low because of these serious risks.

*MRC*: Three RCTs reported MRC scores (figure 3). We observed a significant improvement in the early rehabilitation group (MRC (three trials, n=196; SMD 0.38, 95% CI 0.10 to 0.66, p=0.009; $I^2=0\%$)).

### Cognitive-related outcomes

*Delirium-free days*: Two RCTs reported the incidence of delirium-free days (figure 3). There was no significant increase in this parameter in the early rehabilitation group according to a random effects model (two trials, n=326; SMD −0.02, 95% CI −0.23 to 0.20, p=0.62; $I^2=0\%$). The risk of bias was determined to be serious because all RCTs included a moderate risk of bias, and imprecision was also determined to be serious because of the small number of included patients as per OIS criteria. These serious risks led to the downgrading of the QoE from high to low.

### Mental health-related outcomes

*Incidence of HAS/HADS*: Two RCTs reported the incidence of HAS/HADS (figure 3). These groups did not differ significantly according to a random effects model (two trials, n=92; OR 0.79, 95% CI 0.29 to 2.12, p=0.64; $I^2=0\%$). The risk of bias was determined to be serious because all RCTs included a moderate risk of bias, and imprecision was also determined to be serious because of the small number of included patients. These serious risks led to the downgrading of the QoE from high to low.

### Secondary outcomes (long-term outcomes)

*EQ5D*: Two RCTs reported the EQ5D scores (figure 4). No significant difference was observed between the two groups according to a random effects model (two trials, n=63; SMD 0.11, 95% CI −0.86 to 1.09, p=0.82; $I^2=72\%$). The risk of bias was deemed serious because all RCTs demonstrated a moderate risk of bias, and inconsistency was deemed serious because of severe heterogeneity (p=0.06, $I^2=72\%$). Imprecision was also considered serious because of the small number of included patients. These serious risks led to the downgrading of the QoE from high to very low.

*SF-36 PF*: Two RCTs reported SF-36 PF scale scores (figure 4). However, these scores were evaluated several months later. Moreover, the timing of SF-36 PF evaluation varied greatly among studies (ie, at hospital discharge and 2, 4 and 6 months after enrolment,[32] or at 6 months after hospital discharge).[29] We observed a significant improvement in the early rehabilitation group (SF-36 PF (two trials, n=191; SMD 2.41, 95% CI −0.75 to 5.58, p=0.14; $I^2=98\%$)). However, the different timings of SF-36 PF scale evaluation among the various studies led to significant heterogeneity. Imprecision was also considered serious because of the small number of included patients as per the OIS criteria. Therefore, these serious risks led to the downgrading of the QoE from high to very low.

**Table 1** Characteristics of included studies

| Source | Population | No of patients | | | Age (years) | | APACHE II score | | Intervention | | Timing of first rehabilitation/ standard care | Follow-up | Outcome | | Ref |
| | | Total | Early rehab | Control | Early rehab | Control | Early rehab | Control | Early rehab | Control | | | Short-term (during the hospitalisation) | Long-term (post discharge) | |
| --- | --- | --- | --- | --- | --- | --- | --- | --- | --- | --- | --- | --- | --- | --- | --- |
| Brummel 2014 | Respiratory failure and/or septic, cardiogenic or haemorrhagic shock. | 44 | 22 | 22 | 62 (48–67) | 60 (51–69) | 21.5 (20.0–28.8) | 27.0 (17.5–31.0) | Passive ROM exercises to independent ambulation, guided by the patient's RASS (daily). | Physical therapy (1–2 per week). | Intervention: 1 (1–1), control: 3 (2–6). | 3 months | VFD, DFD | EQ5D | 27 |
| Hodgson 2016 | Critically ill adults mechanically ventilated >24 hours. | 50 | 29 | 21 | 64±12 | 53±15 | 19.8±9.8 | 15.9±6.9 | Functional rehabilitation treatment conducted at the highest level of activity. | Not protocolised and all usual unit practice was continued. | Intervention: 3 (2–4), control: 3 (2–4) | 6 months | MRC, ICU-AW, mechanical ventilation days | EQ5D | 31 |
| Jones 2015 | Patients ≥45 years, had a combined ICU and pre-ICU stay ≥5 days. | 42 | 22 | 20 | 64±13 | 60±12 | 17±10 | 14±4 | In addition to the self-help programme, a 6-week programme of supervised physiotherapy sessions. | Patient-controlled self-help rehabilitation programme. | NA | 3 months | VFD, HAS/HADS | SF-36 PF | 28 |
| Kayambu 2015 | Sepsis patients >18 years who remained mechanically ventilated >48 hour. | 50 | 26 | 24 | 62.5 (30–83) | 65.5 (37–85) | 28±7.6 | 27±6.8 | EMS, passive ROM, active ROM, sitting out of bed, transfers, ambulation for 30 min, one to two times daily. | Standard ICU care, which included physical therapy. | Within 48 hours of sepsis | 6 months | MRC, VFD, HAS/HADS | SF-36 PF | 29 |
| Morris 2016 | Patients with acute respiratory failure requiring MV. | 300 | 150 | 150 | 55±17 | 58±14 | NA | NA | Passive ROM, physical therapy and progressive resistance exercises for 7 days per week. | Physical therapy weekday when ordered by the clinical team. | Intervention: 1 (0–2) Control: 7 (4–10) | 6 months | VFD, DFD | SF-36 PF | 32 |
| Schweickert 2009 | Patients who had been on MV for <72 hour. | 104 | 49 | 55 | 57.7 (36.3–69.1) | 54.4 (46.5–66.4) | 20 (15.8–24) | 19 (13.3–23) | Early exercise and physical and occupational therapy on the day of enrolment. | Therapy as ordered by the primary care team. | Intervention: 1.5 (1.0–2.1) Control: 7.4 (6.0–10.9) | NA | MRC, ICU-AW, VFD, DFD | | 30 |

APACHE II score, Acute Physiology and Chronic Health Evaluation II score; AW, acquired weakness; DFD, delirium-free days; early rehab, early rehabilitation; EMS, electrical muscle stimulation; EQ5D, EuroQol 5 Dimensions; HAS/HADS, Hospital Anxiety and Depression Scale; ICU, intensive care unit; MRC, Medical Research Council; MV, mechanical ventilation; NA, not applicable; RASS, Richmond agitation-sedation scale; ROM, range of motion; SF-36 PF, 36-Item Short Form Health Survey Physical Function scale; VFD, ventilator-free days .

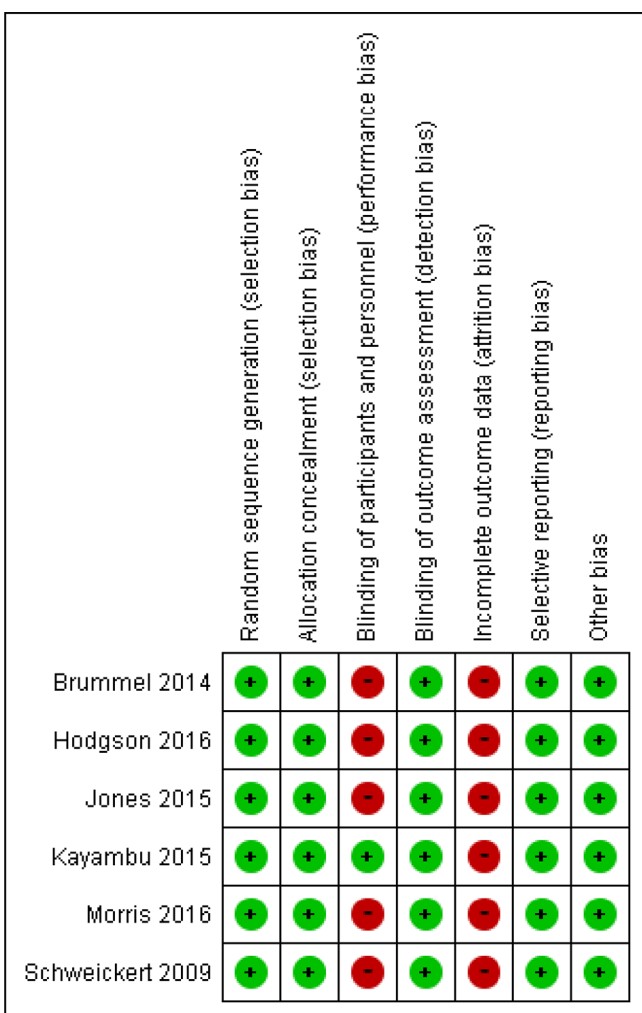

**Figure 2** Risk of bias summary.

## DISCUSSION
### Summary of the main results
This meta-analysis revealed that early rehabilitation significantly improved short-term physical function in patients with critical illness, as assessed by the MRC scoring system and the incidence of ICU-AW. However, it did not significantly improve the patients' cognitive and mental status-related outcomes. These results suggest that early rehabilitation has a limited effect on the prevention of PICS in patients with critical illness, and only reflect improvements in short-term physical function.

### Strengths of the review
To our knowledge, the present work is the first meta-analysis of comprehensive PICS based on RCTs in which the study intervention populations were limited to early rehabilitation. Additionally, we used the GRADE approach in the review process which allowed us to make judgements about the QoE and strength of recommendations in a systematic and transparent manner. Thus, the present meta-analysis allowed us determine whether critical care physicians should initiate early rehabilitation for patients with critical illness, and provided new evidence about

the effectiveness of physiotherapy interventions aimed at early mobilisation.[33]

### Agreements and disagreements with other studies or reviews
Four systematic reviews have addressed the effect of rehabilitation[33–35] or early rehabilitation[36] during an ICU stay on functional status. Through a meta-analysis, Kayambu *et al* found that physical therapy in the ICU may improve muscle strength, physical function, quality of life, ventilator-free days, ICU stay and hospital stay; however, the authors included one RCT involving patients who underwent laparotomy (among 10 total RCTs), and another study initiated physical therapy 30 days after bed rest.[34] Therefore, the results cannot be directly applied to our clinical practice because of the heterogeneity of the study population. The present meta-analysis showed that early rehabilitation did not improve patients' quality of life. The inconsistency of these results might be attributable to differences in the methods used to assess quality of life. Kayambu *et al* used the Medical Outcomes Study 36-Item Short Form survey score, whereas the present review used the EQ5D.

Castro-Avila *et al* conducted a systematic review and meta-analysis of the effect of early rehabilitation during an ICU stay on functional status[34] and included the study reported by Denehy *et al*.[37] However, we did not include this study in our systematic review, as it implemented early active rehabilitation in the control group. This implementation was the main reason underlying the authors' conclusion that early rehabilitation was not associated with improvements in the walking distance and incidence of ICU-AW. Furthermore, Tipping *et al* reported a systematic review regarding the effects of active mobilisation and rehabilitation in the ICU on mortality and function,[35] and included a study by Moss *et al*.[38] However, that study did not initiate rehabilitation during the early phase and therefore was also not included in the current systematic review.

### Clinical implementation
The initiation of early rehabilitation improved short-term outcomes, such as the physical function score and the incidence of ICU-AW in the present systematic review; however, it did not improve. The tendency to favour early rehabilitation during mechanical ventilation suggests that early rehabilitation could potentially improve short-term survival and maintain and increase muscle strength.

However, the initiation of rehabilitation during acute phase may inflict tremendous physical stress and exhaustion on patients in the ICU, thus increasing ICU mortality.[39] Although several studies included in the present meta-analysis described the safety of early physiotherapy in the ICU,[27 29–32 40] only two outcomes related to the safety of early rehabilitation, the blood lactate level[29] and early physiotherapy termination rate,[31] were compared between two groups. Therefore, the study conclusions should be approached with caution.

**Table 2** Quality assessment and summary of findings for the main comparison

| Quality assessment | | | | | | | No of patients | | Effect | | Quality | Importance |
|---|---|---|---|---|---|---|---|---|---|---|---|---|
| No of studies | Study design | Risk of bias | Inconsistency | Indirectness | Imprecision | Other considerations | Early rehabilitation | Control | Relative (95% CI) | Absolute (95% CI) | | |
| **Incidence of ICU-AW** | | | | | | | | | | | | |
| 2 | Randomised trials | Serious* | Not serious | Not serious | Serious† | None | 22/78 (28.2%) | 37/76 (48.7%) | OR 0.42 (0.22 to 0.82) | **202** fewer per **1000** (from 49 fewer to 314 fewer) | ⊕⊕◯◯ Low | Critical |
| **MRC** | | | | | | | | | | | | |
| 3 | Randomised trials | Serious* | Not serious | Not serious | Serious† | None | 97 | 99 | – | SMD 0.38 higher (0.1 higher to 0.66 higher) | ⊕⊕◯◯ Low | Critical |
| **Delirium-free days** | | | | | | | | | | | | |
| 3 | Randomised trials | Serious* | Not serious | Not serious | Serious† | None | 164 | 162 | – | SMD 0.02 lower (0.23 lower to 0.2 higher) | ⊕⊕◯◯ Low | Critical |
| **Incidence of HADS/HAS** | | | | | | | | | | | | |
| 2 | Randomised trials | Serious* | Serious | Not serious | Serious† | None | 10/48 (20.8%) | 11/44 (25.0%) | OR 0.79 (0.29 to 2.12) | **42** fewer per **1000** (from 162 fewer to 164 more) | ⊕◯◯◯ Very low | Critical |
| **Health related QOL score (EQ5D)** | | | | | | | | | | | | |
| 2 | Randomised trials | Serious* | Serious‡ | Not serious | Serious† | None | 35 | 28 | – | SMD 0.11 SD higher (0.86 lower to 1.09 higher) | ⊕◯◯◯ Very low | Critical |
| **SF-36 PF** | | | | | | | | | | | | |
| 2 | Randomised trials | Serious* | Not serious | Not serious | Serious† | None | 93 | 98 | – | SMD 2.41 higher (0.75 lower to 5.58 higher) | ⊕⊕◯◯ Low | Critical |

*Most of the studies are classified as unclear or high risk of bias (performance bias and attrition bias).
†CIs for absolute effects are wide and include no effect and include important harm and/or important benefit and/or small sample size (the number of events is <300).
‡Substantial heterogeneity present.
AW, acquired weakness; EQ5D, EuroQol 5 Dimensions; HADS/HAS, Hospital Anxiety and Depression Scale; ICU, intensive care unit; MRC, Medical Research Council; QOL, quality of life; SF-36 PF, 36-Item Short Form Health Survey Physical Function scale; SMD, standardised mean difference.

## Short-term outcome

## 1. Physical-related outcomes

### A Incidence of ICU-AW

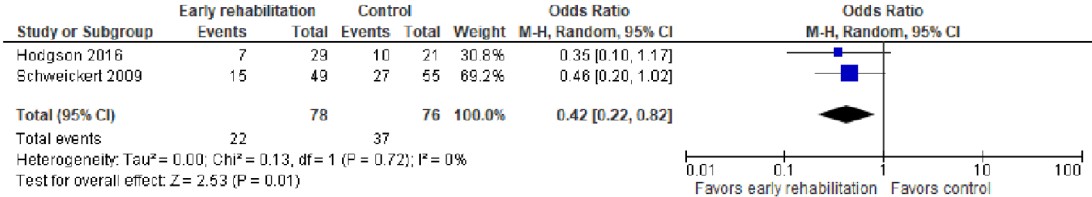

### B MRC

## 2. Cognitive-related outcome

### Delirium-free days

## 3. Mental health-related outcome

### Incidence of HAS/HADS

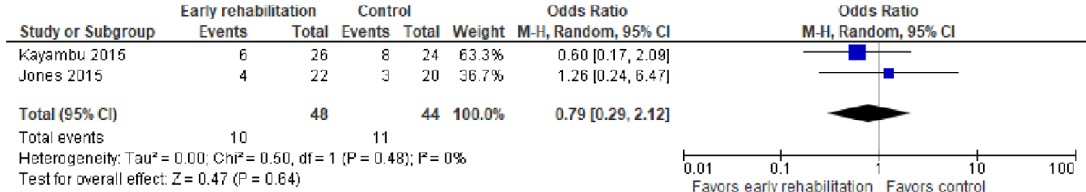

**Figure 3** The effect of early rehabilitation on short-term outcomes in postintensive care syndrome (PICS) in intensive care unit (ICU) patients. (1) Physical-related outcomes (A) Incidence of ICU-acquired weakness (AW). (B) Medical Research Council (MRC) sum score. (2) Cognitive-related outcomes, characterised by delirium-free days. (3) Mental status-related outcomes, characterised by the Hospital Anxiety and Depression Scale (HAS/HADS) score.

### Limitations of the review

The current *meta-analysis* had several limitations. First, the analysis of each outcome included only a small number of patients. Although Schweickert *et al* reported data concerning delirium, we could not extract cognitive-related outcomes because no data about delirium-free days were not addressed directly.[30] Second, the studies included in the current systematic review did not measure the domains of PICS beyond hospital discharge. Third, two of the included studies were pilot/feasibility trials and not powered to identify differences in physical or cognitive outcomes.[27 31] However, it was reasonable to initiate RCTs in the context of feasibility studies because no large multicentre trials of the effects of early mobilisation in

## Long-term outcome

### 1 Health-related QOL scores

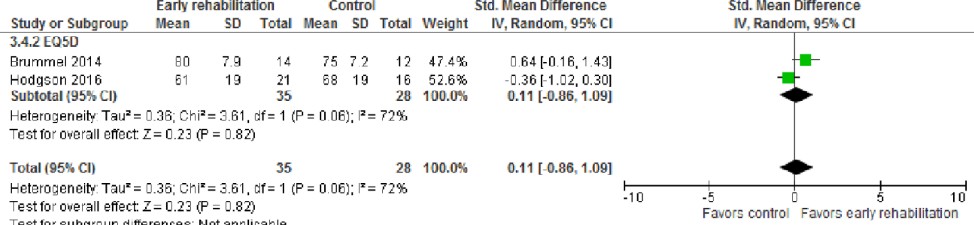

### 2 SF-36PF

**Figure 4** The effect of early rehabilitation on long-term outcomes in postintensive care syndrome (PICS) in intensive care unit (ICU) patients. The effect of early rehabilitation on health-related quality of life (QOL) scores and in ICU patients. (1) Health-related quality of life (QOL) scores calculated from the EuroQol 5 Dimensions (EQ5D). (2) Medical Outcomes Study 36-Item Short Form Health Survey Physical Function scale (SF-36 PF).

the ICU had been published before the study conducted by Brummel *et al*, and little evidence to support the feasibility of individual patient randomisation across multiple sites using a complex intervention (ie, early mobilisation) had been reported before the study conducted by Hodgson *et al*. Moreover, these two RCTs demonstrated the same level of risk as other RCTs, according to risk of bias summary conducted in this meta-analysis (figure 2).

Fourth, we could not confirm the exact first day of rehabilitation initiation in the three studies, although 'early rehabilitation' was described in the text, and we contacted the authors to acquire additional information.[28 40] Fifth, we could not find any data regarding direct cognitive assessment in the selected papers. Although cognitive impairment at hospital discharge may not predict long-term cognitive impairment,[41] Pandharipande *et al* reported that a longer duration of delirium on admission was associated with worse global cognition and executive function scores in the surgical ICU at 3 and 12 months.[42] Therefore, we believe that the assessment of delirium in the ICU and hospital is a very important predictor of long-term cognitive impairment postdischarge. Sixth, as this meta-analysis included studies that had implemented various 'early rehabilitation' protocols, further analyses should include large trials with strict and comparable rehabilitation algorithms adjusted for the initiation time, type and intensity. Despite these several limitations, our meta-analysis clarifies the effectiveness of early rehabilitation for the prevention of PICS in survivors of critical illness.

## CONCLUSION

Early rehabilitation has a limited effect on the prevention of PICS, although it led to significant improvements in short-term physical-related outcomes, including MRC scores and the incidence of ICU-AW. However, early rehabilitation had no significant effect on cognitive function and mental health-related outcomes or mortality in patients with critical illness. Additional large-scale, rigorous RCTs are needed to confirm our results.

**Author affiliations**
[1]Division of Infectious Diseases and Infection Control, Tohoku Medical and Pharmaceutical University, Sendai, Miyagi, Japan
[2]Emergency Medical Center, Kagawa University Hospital, Kita-gun, Kagawa, Japan
[3]Department of Emergency and Critical Care Medicine, Graduate School of Medicine, University of the Ryukyus, Nakagami-gun, Okinawa, Japan
[4]Department of Intensive Care Medicine, Yokohama City Minato Red Cross Hospital, Yokohama, Kanagawa, Japan
[5]Division of Trauma and Surgical Critical Care, Osaka General Medical Center, Osaka City, Osaka, Japan
[6]Department of Emergency and Critical Care Medicine, Tokai University Hachioji Hospital, Hachioji, Tokyo, Japan
[7]Department of Anaesthesiology and Critical Care Medicine, Fujita Health University School of Medicine, Toyoake, Aichi, Japan

**Contributors** YK, RF, TH, JH, TT and SI conceived the idea for this systematic review. YK, RF, TH, JH, TT and SI developed the methodology for the systematic review, and KY and ON supervised the methodological process. The manuscript was drafted by YK and SI. RF, TH, JH, TT and KY, and revised by ON. All authors critically reviewed and approved the final manuscript.

**Funding** This research received no specific grant from any funding agency in the public, commercial or not-for-profit sectors.

**Competing interests** None declared.

**Patient consent** Not required.

**Provenance and peer review** Not commissioned; externally peer reviewed.

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
