## [Reviewer comments · BMJ Open]

ARTICLE DETAILS

TITLE (PROVISIONAL)	Early rehabilitation to prevent post-intensive care syndrome in critically ill patients: a systematic review and meta-analysis
AUTHORS	Fuke, Ryota; Hifumi, Toru; Kondo, Yutaka; Hatakeyama, Junji; Takei, Tetsuhiro; Yamakawa, Kazuma; Inoue, Shigeaki; Nishida, Osamu

VERSION 1 – REVIEW

REVIEWER	Maria Kompoti Intensive Care Unit, Thriassion Hospital of Eleusis, Athens, Greece
REVIEW RETURNED	20-Oct-2017

GENERAL COMMENTS	As I have previously mentioned, I have no objection that this systematic review has been conducted following a robust methodology. Far from intending to "contest the methodology and design of the study", I would expect the authors to comment on the wide scale of "early rehabilitation" protocols implemented and the need of large trials with strict and comparable rehabilitation algorithms, which would yield safer conclusions. Submitting a well-designed protocol is one step. Encountering real data is another and analysis (especially meta-analysis) should not be a mechanistic process which ignores evident limitations. I have no intention to further pursue this matter, but would expect some reference to the wide range of "early rehabilitation" protocols and the need of trials with more strictly defined protocols, which would result in safer conclusions.
--

REVIEWER	Dr Geetha Kayambu National University Hospital, Singapore
REVIEW RETURNED	24-Oct-2017

GENERAL COMMENTS	This "review" is still referred to as a study in certain sections of the paper e,g Page 4 and 16. I still urge the authors to fix this thoroughly. The authors have managed to answer and edit the paper according to reviewer comments which has improved the state of the manuscript.
---

REVIEWER	Matthew J Page Monash University, Australia
REVIEW RETURNED	03-Nov-2017

GENERAL COMMENTS	The authors stated that they used the random-effects for meta-analyses, but have not stated which between-study variance estimator was used in this model. There is also no justification for the choice of model selected. The authors state in the Methods section that they investigated publication bias using a funnel plot, and tested for funnel plot asymmetry using the Egger test. However, no funnel plot is presented in the review, and the results of the Egger test are not referred to in the Results section, so I am unsure if these analyses were even undertaken. Also, there is clear guidance in several sources (e.g. https://www.ncbi.nlm.nih.gov/pubmed/21784880) stating that funnel plot asymmetry tests should only be used when a minimum of 10 studies of varying size are included. Yet the largest number of studies included in the meta-analyses in this review was three. The authors stated that “We performed our analysis based on all published data or data made available to us”, and cite the following article to justify this statement: Duan EH, Oczkowski SJ, Belley-Cote E, et al. beta-Blockers in sepsis: protocol for a systematic review and meta-analysis of randomised control trials. BMJ open 2016;6(6):e012466. It is unclear why Duan et al. have been cited. It is also unclear which data were made available to the authors, so a footnote under each forest plot describing which data were published and which were supplied would help. For the MRC, delirium-free days, EQ5D, SF-36 PF outcomes, it is stated in the main text that the results are presented in “ln(OR)” metrics, but in the Summary of Findings tables and forest plots generated in RevMan, it is stated that the effect estimates for these four outcomes are standardised mean differences (SMDs). I suggest the authors explain this discrepancy, and consistently report the effect estimates across the different sections of the review. In Figure 4, a meta-analysis of SF-36 PF scores is presented. The standard deviations of both groups in Morris 2016 are remarkably small. I suggest the authors check that they have not entered standard errors incorrectly.
--

REVIEWER	Ian Fyffe Simon Fraser University, Canada
REVIEW RETURNED	12-Nov-2017

GENERAL COMMENTS	I appreciated the rigorous approach to this meta-analysis. By this I mean the utilization of the GRADE system as well as the PRISMA guidelines. Furthermore, this area is an important field of research. For a meta-analysis, it is often stated that nine studies are a good rigorous number to include and that the minimum number that can be included is two. For specific primary outcomes: MRC relied upon three RCTs, physical-related outcomes used two, cognitive-related outcomes two, mental-health outcomes two, EQ5D two, and SF-36 PF two. This is worrisome. Currently, I believe that there is a lack of information regarding the individual RCTs that have been included. Since the majority of
---

	outcomes only utilize two RCTs, it is extremely important that the RCTs that have been selected are adequately powered and as free from methodological shortcomings as possible. I would like to see more information regarding the sample size for each individual RCT as well as the strengths of each. Currently, the way this has been reported in the paper is by giving the total n size for all trials together. Unfortunately the reader has no way of knowing the n size of individual RCTs. This could be problematic if one or both of the two selected RCTs are under powered. This was mentioned in the limitations section, when it was written that: "two of the included studies were pilot/feasibility trials and not powered to find differences in physical or cognitive outcomes". I believe that this paper could be stronger and that this limitation could be addressed if more information was included from page 10-13. From a statistical perspective this paper would come across as more transparent if additional information was provided on the individual RCTs. This could include an explanation as to why they were selected from a statistical and methodological perspective. This is particularly important because many of the outcomes only meet the minimum level of studies for a meta-analysis. If statistical power is an issue regarding individual outcomes, then this should be addressed more clearly. This paper has been thoughtfully constructed and includes important information. I believe that it should have revisions, however. This is because a reader has no way of knowing if conclusions have been made on individual outcomes based on studies that are under powered or inadequate for inclusion in a meta-analysis with so few included studies.
--	--

VERSION 1 – AUTHOR RESPONSE

Reviewers' Comments to Author:

Reviewer: 1

Reviewer Name: Maria Kompoti

Institution and Country: Intensive Care Unit, Thriassion Hospital of Eleusis, Athens, Greece

Competing Interests: None declared.

As I have previously mentioned, I have no objection that this systematic review has been conducted following a robust methodology. Far from intending to "contest the methodology and design of the study", I would expect the authors to comment on the wide scale of "early rehabilitation" protocols implemented and the need of large trials with strict and comparable rehabilitation algorithms, which would yield safer conclusions. Submitting a well-designed protocol is one step. Encountering real data is another and analysis (especially meta-analysis) should not be a mechanistic process which ignores evident limitations.

I have no intention to further pursue this matter, but would expect some reference to the wide range of "early rehabilitation" protocols and the need of trials with more strictly defined protocols, which would result in safer conclusions.

Reply: Thank you for your valuable comments. We have added the following sentences to the section describing limitations:

Sixth, as this meta-analysis included studies that had implemented various 'early rehabilitation' protocols, further analyses should include large trials with strict and comparable rehabilitation algorithms adjusted for the initiation time, type, and intensity.

Reviewer: 2

Reviewer Name: Dr Geetha Kayambu

Institution and Country: National University Hospital, Singapore

Competing Interests: None declared

This "review" is still referred to as a study in certain sections of the paper e.g Page 4 and 16. I still urge the authors to fix this thoroughly.

The authors have managed to answer and edit the paper according to reviewer comments which has improved the state of the manuscript.

Reply: Thank you for your valuable comments. We have changed the sentences in accordance with your advice.

Strengths and limitations of this meta-analysis study

Limitations of the review

The current meta-analysis study had several limitations.

Reviewer: 3

Reviewer Name: Matthew J Page

Institution and Country: Monash University, Australia

Competing Interests: None declared

The authors stated that they used the random-effects for meta-analyses, but have not stated which between-study variance estimator was used in this model. There is also no justification for the choice of model selected.

Reply: Thank you for your valuable comments. We have described "Estimates were pooled using a random effects model" in Data synthesis in Method section.

The authors state in the Methods section that they investigated publication bias using a funnel plot, and tested for funnel plot asymmetry using the Egger test. However, no funnel plot is presented in the review, and the results of the Egger test are not referred to in the Results section, so I am unsure if these analyses were even undertaken. Also, there is clear guidance in several sources (e.g. <https://www.ncbi.nlm.nih.gov/pubmed/21784880>) stating that funnel plot asymmetry tests should only be used when a minimum of 10 studies of varying size are included. Yet the largest number of studies included in the meta-analyses in this review was three.

Reply: Thank you for your valuable comments. We have deleted the following sentences in accordance with your advice:

Assessment of reporting bias

We investigated the possibility of publication bias using a funnel plot. To test for funnel plot asymmetry, we used the Egger test using STATA SE Statistical Software (Release 13. College Station, TX: StataCorp LP) 27 28.

The authors stated that "We performed our analysis based on all published data or data made available to us", and cite the following article to justify this statement: Duan EH, Oczkowski SJ, Belley-Cote E, et al. beta-Blockers in sepsis: protocol for a systematic review and meta-analysis of randomised control trials. *BMJ open* 2016;6(6):e012466. It is unclear why Duan et al. have been

cited. It is also unclear which data were made available to the authors, so a footnote under each forest plot describing which data were published and which were supplied would help.

Reply: Thank you for your valuable comments. We have deleted the reference as shown below: We performed our analysis based on all published or available data 25.

For the MRC, delirium-free days, EQ5D, SF-36 PF outcomes, it is stated in the main text that the results are presented in "ln(OR)" metrics, but in the Summary of Findings tables and forest plots generated in RevMan, it is stated that the effect estimates for these four outcomes are standardised mean differences (SMDs). I suggest the authors explain this discrepancy, and consistently report the effect estimates across the different sections of the review.

Reply: Thank you for your valuable comments. Despite changes in the contents of the analysis, we did not correct the effect estimates at the time of the last posting. We have corrected 'ln(OR)' to 'SMD'.

In Figure 4, a meta-analysis of SF-36 PF scores is presented. The standard deviations of both groups in Morris 2016 are remarkably small. I suggest the authors check that they have not entered standard errors incorrectly.

Reply: Thank you for your valuable comments. We re-evaluated the two SDs and confirmed that they were correct.

Reviewer: 4

Reviewer Name: Ian Fyffe

Institution and Country: Simon Fraser University, Canada

Competing Interests: None declared

I appreciated the rigorous approach to this meta-analysis. By this I mean the utilization of the GRADE system as well as the PRISMA guidelines. Furthermore, this area is an important field of research. For a meta-analysis, it is often stated that nine studies are a good rigorous number to include and that the minimum number that can be included is two. For specific primary outcomes: MRC relied upon three RCTs, physical-related outcomes used two, cognitive-related outcomes two, mental-health outcomes two, EQ5D two, and SF-36 PF two. This is worrisome.

Currently, I believe that there is a lack of information regarding the individual RCTs that have been included. Since the majority of outcomes only utilize two RCTs, it is extremely important that the RCTs that have been selected are adequately powered and as free from methodological shortcomings as possible. I would like to see more information regarding the sample size for each individual RCT as well as the strengths of each. Currently, the way this has been reported in the paper is by giving the total n size for all trials together. Unfortunately the reader has no way of knowing the n size of individual RCTs. This could be problematic if one or both of the two selected RCTs are under powered.

Reply: Thank you for your valuable comments. We have described the number of patients in each RCT in both the Figures and Table 1.

This was mentioned in the limitations section, when it was written that: "two of the included studies were pilot/feasibility trials and not powered to find differences in physical or cognitive outcomes". I believe that this paper could be stronger and that this limitation could be addressed if more information was included from page 10-13. From a statistical perspective this paper would come

across as more transparent if additional information was provided on the individual RCTs. This could include an explanation as to why they were selected from a statistical and methodological perspective. This is particularly important because many of the outcomes only meet the minimum level of studies for a meta-analysis. If statistical power is an issue regarding individual outcomes, then this should be addressed more clearly.

Reply: Thank you for your valuable comments. We have added the following sentences regarding limitations, per your recommendation.

Third, two of the included studies were pilot/feasibility trials and not powered to identify differences in physical or cognitive outcomes 27-31. However, it was reasonable to initiate RCTs in the context of feasibility studies because no large multicentre trials of the effects of early mobilisation in the ICU had been published before the study conducted by Brummel et al., and little evidence to support the feasibility of individual patient randomisation across multiple sites using a complex intervention (i.e. early mobilisation) had been reported before the study conducted by Hodgson et al. Moreover, these two RCTs demonstrated the same level of risk as other RCTs, according to risk of bias summary conducted in this meta-analysis (Figure 2).

This paper has been thoughtfully constructed and includes important information. I believe that it should have revisions, however. This is because a reader has no way of knowing if conclusions have been made on individual outcomes based on studies that are under powered or inadequate for inclusion in a meta-analysis with so few included studies.

VERSION 2 – REVIEW

REVIEWER	Matthew J Page Monash University
REVIEW RETURNED	11-Dec-2017
GENERAL COMMENTS	Thank you for addressing all comments

VERSION 2 – AUTHOR RESPONSE

Editors' Comments to Author:

- Can you please improve the formatting/ presentation of table 1? Currently the columns are too narrow making this table very difficult to read.

Reply :

We have simplified descriptions using abbreviations, and double-spaced in Table 1.

- Please revise the 'strengths and limitations' section on page 4, which is currently not following journal guidelines. Each bullet point should be a separate strength or limitation. It should not be a summary of the study and its findings so the first and third bullet points need replacing. As a reminder, this section should contain up to five short bullet points, no longer than one sentence each, that relate specifically to the methods or design of the study reported (see: <http://bmjopen.bmj.com/site/about/guidelines.xhtml#articletypes>).

Reply :

We have revised the 'strengths and limitations' section as followings;

Strengths and limitations of this meta-analysis

- This meta-analysis is the first meta-analysis of comprehensive PICS based on RCTs in which the study intervention populations were limited to early rehabilitation.
- This meta-analysis was mainly limited by (1) the small number of patients in the included RCTs; (2) a lack of detailed analysis of the adverse effects and PICS domains after hospital discharge; and (3) no confirmation of the exact first day of rehabilitation initiation in three of the eight included RCTs.
- We used the GRADE approach in the review process.

- The literature search goes up to June 2016, which is more than 12 months ago now. Can you please update your search to check that there has not been any recent studies published that meet the inclusion criteria?

Reply : Thank you for your comments. We have searched the Medline (via PubMed from 8 June 2016 to 15 January 2018) using same key search terms. A total of 320 citations were identified, but no studies were judged to be potentially eligible based on the inclusion criteria.